# Surgical Treatment for Advanced Oropharyngeal Cancer: A Narrative Review

**DOI:** 10.3390/medicina59020304

**Published:** 2023-02-07

**Authors:** Antonino Maniaci, Sheng-Po Hao, Francesco Cancemi, Damiano Giardini, Emanuele Checcoli, Francesco Soprani, Giannicola Iannella, Claudio Vicini, Salvatore Cocuzza, Ignazio La Mantia, Nicolas Fakhry, Andrea De Vito

**Affiliations:** 1Department of Medical, Surgical and Advanced Technologies G.F.Ingrassia, ENT Section, University of Catania, 95123 Catania, Italy; 2Faculté des Sciences Médicales et Paramédicales, Aix-Marseille Université, 13005 Marseille, France; 3 Pôle PROMO, Service ORL et Chirurgie Cervico-Faciale, Hôpital de la Conception, Assistance Publique des Hôpitaux de Marseille, 13005 Marseille, France; 4Department of Otolaryngology Head and Neck Surgery, Shin Kong Wu Ho-Su Memorial Hospital, School of Medicine, Fu-Jen University, Taipei 100, Taiwan; 5Department of Surgery, ENT Unit, “Santa Maria delle Croci” Hospital, 48121 Ravenna, Italy; 6“Umberto I” Hospital, Health Local Agency of Romagna, 48022 Lugo, Italy; 7Department of Surgery, ENT Unit, “Morgagni-Pierantoni” Hospital, 47121 Forlì, Italy; 8“Degli Infermi” Hospital, Health Local Agency of Romagna, 48018 Faenza, Italy

**Keywords:** oropharyngeal cancer, HPV, transoral robotic surgery, tonsil cancer, base of tongue cancer

## Abstract

*Background and Objectives*: to describe current scientific knowledge regarding the treatment options in advanced oropharyngeal cancer. The standard care for advanced oropharyngeal cancer (OPSCC) has been chemoradiotherapy, although surgical approaches followed by adjuvant treatment have been proposed. The best therapy for each patient should be decided by an interdisciplinary tumour-board. Different strategies should be considered for the specific patient’s treatment: surgery, chemotherapy and radiation therapy or combinations of them. The treatment choice is influenced by tumour variability and prognostic factors, but it also depends on cancer extension, extranodal extension, nervous invasion, human papilloma virus (HPV) presence, making the decisional algorithm not always clear. HPV-related OPSCC is strongly associated with a favourable overall survival (OS) and disease-free survival rate (DSS); by contrast, HPV-negative OPSCC often flags a worse prognosis. Consequently, the American Joint Committee on Cancer (AJCC) differentiates OPSCC treatment and prognosis based on HPV status. *Methods*: we carried out a review of current scientific literature to analyze the different indications and limitations of surgical treatment options in OPSCC stage III and IV. *Conclusion*: robotic surgery or open approaches with reconstructive flaps can be considered in advanced stages, resulting in the de-intensification of subsequent systemic therapy and fewer related side effects. Furthermore, in the event of the primary failure of systemic therapy or disease recurrence, the surgical approach constitutes an additional therapeutic option which lengthens patient survival functions.

## 1. Introduction

Head and neck cancer is widespread throughout the world. According to the latest global cancer incidence and mortality reports, the estimated cancer death rate is 51 per 100,000 in the population and up to 80% of new diagnoses are already in an advanced stage [1,2].

Among all cancers, oropharyngeal squamous cell carcinoma (OPSCC) has one of the most rapidly rising incidences in high-income countries [3,4]. The American Joint Committee on Cancer (AJCC) differentiates OPSCC prognosis based on human papilloma virus (HPV) p16+ status [3], introducing a different staging category in the latest eighth edition of the AJCC TNM staging system [1,2,3,4,5,6,7,8,9,10,11,12,13,14,15,16,17]. P16-related OPSCC is strongly associated with a favourable overall survival (OS) and disease-free survival rate (DSS), while, conversely, p16 negative OPSCC often flags a worse prognosis [1,2,3,4,5,6,7,8,9,10,11,12,13,14,15,16,17]. HPV positivity on immunochemistry is not the same as p16 +, considering that only 80% of p16+ are also HPV-positive [6,7,8,9,10,11,12,13,14,15]. Several authors have reported significantly worse survival patterns of OPSCC p16-negative patients, despite higher nodal metastasis rates of p-16 positive cancers [7,8]. The Radiation Therapy Oncology Group (RTOG) trials performed by Ang et al., in 2010, analyzed survival risk functions among OPSCC patients with stage III/IV cancer [9]. Better 3-year rates were found in HPV-positive subjects compared to HPV-negative subjects (82.4%, vs. 57.1% among patients; *p* < 0.001 at log-rank test). In 2021, the percentage of HPV-related OPSCC was reported to be globally the 33% of total OPSCC. However, the percentage varies considerably among different regions, and socio-economic status, ranging from 0% in southern India to 85% in Lebanon [10]. Different OPSCC anatomic subsites were also hypothesized as a risk stratification factor for patient survival, especially between those involving tonsillar related areas (TRA) vs. non-tonsillar (nTRA) ones [11,12,13]. Recently, Tham et al. found OPSCC subsites to be an independent prognostic factor for survival (TRA vs. nTRA HR: 0.76, 95% CI: 0.67–0.86, *p* < 0.0001) [12]. Furthermore, in 2015 Iyer et al. reported poorer survival rates in p16-positive soft palate tumours compared to those on the base of the tongue and tonsil (RR 4.8, CI 1.3–17.2; *p* = 0.016) [8,14,15,16,17]. 

The literature suggested that the standard care for advanced OPSCC is chemoradiotherapy, although surgical approaches followed by adjuvant treatment have been proposed [8,9,10,11,12,13,14,15,16,17,18,19,20,21,22,23]. Several studies in the literature have reported comparable 3-year overall survival (OS) rates in the surgical and chemoradiotherapy treatment of advanced OPSCC (ranging from 40% to 80% in both approaches) [19,20,21,22]. Indeed, some authors suggested that extensive patient phenotypic variability, often conditioned by factors such as HPV status, p16, staging, grading or extranodal extension (ENE) positivity, raise the need for a more critical literature interpretation, especially regarding the treatment response in advanced stage tumours [1,2,3,4,5,6,7,8,9,10,11,12,13,14,15,16,17,18].

The clinical diagnosis and staging of OPSCC is based on the endoscopic and the imaging assessment of aerodigestive tract. The anatomy of the oro-hypopharyngeal airway allows direct clinical visualization and conventional white light fiber-optic naso-endoscopy aids in the morphologic examination of a visible lesion. However, early staged lesions can still be missed, especially because they are frequently asymptomatic in the majority of OPSCC patients.

Narrow band imaging (NBI) technology uses the application of specific light filters (blue and green light at wavelengths 415 and 540 nm, respectively) which, absorbed by haemoglobin, allows enhancement of the capillary network of the upperaerodigestive tract. Considering that angiogenesis represents one of the earliest changes in the carcinogenic process, NBI can thus differentiate between normal tissue, dysplasia, and malignancy, with greater diagnostic accuracy, compared to conventional upper airway assessment techniques [23,24,25,26,27,28,29].

Specific literature data have confirmed the efficacy of NBI in analysing different clinical lesions, from suspicious premalignant to malignant lesions, including leucoplakia and erythroplakia, to chronic nonhealing ulcers [30,31,32,33].

The main NBI patterns described for OPSCC are well-demarcated areas and irregular microvascular patterns, which consist of the presence of brownish dots with extension, dilatation, elongation, weaving and differing shapes in the intraepithelial layer and are related to interpapillary capillary loop (IPCL). These patterns are related to neoangiogenesis process and could be consistent with pre-neoplastic or neoplastic lesions. Furthermore, literature data reported high sensitivity and specificity of the NBI in the detection of early mucosal cancer of the oropharynx and hypopharynx as well as in the surveillance of cancer of the head and neck [29,30,31,32,33,34].

NBI could represent a useful tool in the intra-operative assessment margins in OPSCC lesions, in addition to piecemeal resection. This is of utmost importance if trans-oral surgical techniques are applied, such as CO2 laser or robotic resection [32,35,36].

Meccariello et al. retrospectively evaluated 58 biopsy-proven OPSCC patients who underwent TORS procedures. In the group who underwent TORS and intra-operative NBI evaluation, frozen section analysis of margins on surgical specimens showed a higher rate of negative superficial lateral margins in the NBI-TORS group compared with the white light-TORS group (87.9% vs. 57.9%, respectively, *p* = 0.02). The sensitivity and specificity of intra-operative use of NBI, respectively, were 72.5% and 66.7% with a negative predictive value of 87.9%. Tumour margin enhancement provided by NBI associated with magnification and a three-dimensional view of the surgical field might increase the capability to achieve an oncologically-safe resection in challenging anatomical areas where minimal curative resection is strongly recommended for function preservation [36].

Recently, NBI has been coupled with enhanced contact endoscopy (ECE), which identifies five vascular patterns, ranging from normal to squamous cell carcinoma, passing through inflammation, hyperplasia, and dysplasia [37]. Carta et al. analysed the sensitivity, specificity, positive predictive value and negative predictive value of ECE in the differentiation between a healthy mucosa and inflammation versus pathologic hyperplasia, dysplasia, and carcinoma, reporting, respectively, 96.6, 93.3, 98.2, 87.5, and 95.9%. Sensitivity and specificity were 100% in differentiation between non-malignant lesions versus squamous cell carcinoma. This preliminary experience reported the accuracy of ECE in the diagnosis of precancerous lesions and squamous cell carcinoma of the oral cavity and oropharynx [37].

Imaging in OPSCC plays a central role in tumor staging and therapeutic planning. Important information on primary tumor location, oropharyngeal subsites involvement, tumor volume, lymph node status and distant metastasis are gathered [38]. In addition, imaging is also fundamental for post-treatment follow-up. Current NCCN guidelines report that OPSCC patients should be studied with head and neck contrast enhanced CT and/or MRI. The selection of CT vs. MRI depends on the availability, patient tolerance for the imaging examination and costs [39,40]. Enhanced CT is the most commonly available modality of oropharyngeal imaging, even though it is limited by dental filling artifact and poor soft tissue contrast resolution. Enhanced MRI reports a superior soft tissue contrast resolution compared to CT but can be limited by significant motion artifact [40].

The detection of base of the tongue (BOT) OPSCC can be difficult due to the tongue’s dense musculature and lack of fat planes. Moreover, lingual tonsils size can be highly variable and therefore increasing evaluation difficulty. BOT OPSCC often originates as a clinically silent modality and spreads laterally to the palatine tonsils, anteriorly to oral tongue portion and posteriorly to the valleculae area. Imaging evaluation of BOT OPSCC needs to include the analysis of tumour extension (submucosal and tongue intrinsic muscles involvement, tumour crossing of the BOT midline, spreading of the pre-epiglottic fat and bone involvement) for a complete tumor staging and therapeutic planning. Literature data reports that enhanced CT is complementary to MRI in analysis BOT OPSCC [41,42].

Imaging evaluation of tonsillar OPSCC primarily involves the assessment of submucosal invasion because there are multiple unobstructed routes through which the tumor may spread into the nasopharynx, parapharyngeal space, masticator space, skull base, and BOT. Enhanced CT is usually the choice for the initial imaging study; however, tonsillar OPSCC, as with SCC of the retromolar trigone, is more accurately studied by MR imaging for its complete evaluation of soft-tissue extension. Evaluation of a tonsillar SCC should include a determination of the extent of submucosal extension, involvement of the pterygoid muscles, extension along the pterygomandibular raphe to the skull base, osseous involvement, and involvement of the cervical lymph nodes [41,42].

OPSCC represents one of the most common sites of unknown head and neck carcinoma. The first clinical presentation is of nodal metastasis. If clinical evaluation and imaging fails to identify the primary tumor site, PET/CT should be ordered before performing FNA, biopsies, and tonsillectomy [43,44].

In OPSCC HPV-positive versus HPV-negative patients, PET/CT imaging results show a significantly higher metabolic rate in HPV-negative compared to HPV-positive patients, and a statistically significantly larger SUVmax, SUVpeak and SUVmean value. Morphological and glycolytic indices of nodal metastases are also overall larger in HPV-positive than in HPV-negative OPSCC [43,44].

An up-to-date review of literature was carried out to analyze different indications of the primary surgical approach, or salvage options after previous systemic treatment failure. We present the following analysis in accordance with the Narrative Review reporting checklist.

## 2. Materials and Methods

### 2.1. Study Searching Protocol

We performed a review of current literature by analysing PubMed, Scopus and Web of Science electronic databases of the last 20 years of literature (from April 2001, to April 2022) by two different authors. Keywords used for the study research were: “Oropharynx carcinoma” and/or “oropharyngeal squamous cell carcinomas” and/or “TORS and Oropharynx” and/or “oropharyngeal squamous cell carcinomas treatment”. The investigators examined titles and abstracts of papers available in English.

### 2.2. Inclusion/Exclusion Criteria

Two independent reviewers (A.M. and G.I.) initially screened all articles by title and abstract; the authors then independently assessed the full-text versions of each publication and excluded those whose content was judged not to be strictly related to the subject of this review.

Exclusion criteria for the study were as follows: (1) studies not in English language; (2) case reports, conference abstracts and letters to the editor; (3) studies with unclear and/or incomplete data; (4) experimental/trial studies or non-clinical studies. 

All studies identified through this type of research have been reviewed and considered for the preparation of this narrative review.

## 3. Results

### 3.1. Overall Surgical Strategies

The current literature confirmed that surgical approaches have limited indications in advanced OPSCC (stage 1II and IV) (Figure 1). 

Surgery should be considered when radio-chemotherapy fails or it is not feasible due to patient choice [13,14,15,16,17,18,19,20,21,22,45,46,47], whereas the main benefits of surgery may be focused on OPSCC early-stages (stage 1 and II) [14,15,16,17,18].

Liederbach et al. [47] performed a contemporary analysis of overall surgical trends in the treatment of OPSCC from 1998 to 2012: the use of surgery decreased from 41.4% in 1998 to 30.4% in 2009 (*p* < 0.001). The surgical trends reversed with an overall increasing of 34.8% reported in AJCC 2012 edition, even if with consistent variation at different T stages: 80.2% of stage I patients receiving surgery compared with 54.0% of stage II patients, 36.8% of stage III patients, and 28.5% of stage IV patients (*p* < 0.001) (Figure 2). 

The surgical techniques applied in advanced OPSCC stages have also changed over time [45]. Initially, open techniques were the primary surgical choice [46]. However, invasive approaches such as mandibulotomy or trans-cervical pharyngotomy resulted in severe functional and aesthetic morbidity [47]. 

Recently, trans-oral surgical techniques have been introduced, reporting excellent functional outcomes without compromising the oncological outcomes, when compared to open surgical procedures [48,49,50]. The introduction of laser CO_2_ and transoral robotic surgery (TORS) approaches resulted as new and remarkable alternatives to chemoradiotherapy, especially in stage I and II oropharyngeal carcinomas [51]. Furthermore, an awareness of better survival patterns amongst HPV-related OPSCC patients boosted interest in systemic treatment de-escalation following mini-invasive trans-oral techniques [52]. However, literature data on trans-oral surgery in advanced stages is limited, due to difficult cancer exposure and possible insufficient surgical radicality with positive resection margins [53,54,55].

Trans-oral surgery is not free of complications, especially if there is involvement of the internal carotid artery, or both lingual arteries [56]. Limited mouth opening does not enable a correct exposure or to reach deep planes up to the prevertebral fascia, constituting a further limitation of this method [57].

Nevertheless, an open surgical approach can play a role in salvage surgery for recurrent OPSCC, in order to reduce morbidity and decrease systemic adjuvant treatment [58].

### 3.2. Primary Surgery for HPV/p16 Positive 

The favorable HPV-related survival rate in OPSCC and continuously increasing incidence induced the AJCC Staging Manual to establish treatment distinction according to p16 status [59].

Primary tumour resection and neck dissection could be considered in HPV-positive patients, despite stage T3-T4 cancer and cN2-3 neck disease [60].

According to the NCCN guidelines, in the case of T1-2 N0-1 tumours, it is possible to perform a surgical resection of the tumour and a single or bilateral neck dissection (II–IV selective neck dissection). If, after surgery, there are no negative prognostic risk factors (extranodal extension, positive margins, close margins, pT3 or pT4 primary, one positive node > 3 cm or multiple positive nodes, nodal disease in levels IV or V, perineural invasion, vascular invasion, lymphatic invasion), it is possible to proceed with a closed follow-up, without adjuvant treatment. In T3-4, N2-3 or patients with negative prognostic factors, adjuvant treatment (radiotherapy alone or radio-chemotherapy) must be performed [61].

Several authors agree with the role of surgery in patients with HPV-positive OPSCC, as it enables better comparable results and lower adjuvant radiation doses compared to primary chemoradiotherapy (CRT) treatments [17,62,63,64].

Furthermore, in a retrospective cohort study, 62 T4a-T4b patients treated with surgery reported significantly better OS, disease specific survival (DSS) and disease-free survival (DFS) in Kaplan–Meier compared to non-surgical treatments protocol (*p* = 0.007, *p* = 0.003, *p* = 0.005, respectively) [22].

Moreover, the trans-oral technique in p16 positive OPSCC patients can also be supported in case of access constraints due to tumour extension, by a combined approach with a pharyngectomy limited to the tumour [65].

### 3.3. Primary Surgery for HPV/p16 Negative

Several authors reported a significantly worse prognosis of OPSCC HPV-negative than OPSCC HPV-positive patients [66,67,68].

Consequently, a separate staging system was presented in the latest 8th AJCC edition for head and neck cancer [69]. Surgical resection is possible in all T3-4, N0-3. If negative prognostic characteristics persist after surgery, it is possible to perform adjuvant radiotherapy alone; if negative features are present, adjuvant radiochemotherapy is recommended.

Recently, a seven multi-centre retrospective study including 474 p-16 negative patients, compared the role of primary surgery with adjuvant radiochemotherapy in advanced-OPSCC stage disease, reporting more favorable prognostic impact in 138 (37%) patients who underwent primary surgery, compared with 233 (63%) non-surgical subjects, with five-year OS, disease-specific survival (DSS) and recurrence-free survival of 76.5, 81.3 and 61.3%, respectively, in the surgical group and 49.9, 61.8 and 43.4%, respectively, in the non-surgical group [70].

A retrospective cohort study on T4a or T4b OPSCC patients found that Kaplan–Meier OS and DSS rates were higher in the p16-negative surgical group than in the non-surgical one; however, statistical significance was not achieved (log-rank *p* = 0.10; log-rank *p* = 0.15, respectively) [22].

Other authors reported high control rates of HPV-negative tumours treated by adjuvant radiation after TORS, ranging from 80 to 90% [55,71,72,73,74,75,76,77,78,79]. De Almeida et al. 2015 analyzed the role of postoperative adjuvant radiation after TORS in 364 OPSCC patients, of which 197 (54.1%) in N2-N3 nodal stage [80]. The authors described a 2-year locoregional control for HPV and p-16 negative subjects (67 vs. 58, respectively), not significantly different with positive ones (*p* = 0.43; *p* = 0.06 respectively) [77].

However, it should be noted that only a few authors have carried out prospective observational studies in HPV and p-16 negative OPSCC [55,80].

Additionally, advanced HPV-negative patients frequently report higher recurrence rates (about 50%), and in this regard, surgery options could also play a salvage role [81,82].

### 3.4. De-Intensification for Adjuvant Treatment

The requirement for de-intensification approaches has been expressed by several authors, especially in the case of OPSCC p16 + patients, of a younger age and with no comorbidities [83,84]. Approaches have aimed to reduce treatment toxicity while preserving disease control and comparable survival results; however, to date, few trials have been completed and many are still ongoing.

Alternative treatment approaches based on surgical de-intensification include neoadjuvant chemotherapy, surgery with adjuvant radiotherapy alone, or with chemoradiotherapy dose-reduction [85,86,87].

In a meta-analysis by Petrelli et al. in 2022, the role of the de-intensification was analyzed in both curative and adjuvant setting. Although the effectiveness in the curative setting had a negative impact on OS as primary outcome (HR = 1.44, 95% CI 1.26–1.66; *p* < 0.01), progression-free survival (PFS), locoregional control (LCR) and distant metastasis (DM) [HR = 2.4, (95% CI 1.84–3.13); HR = 2.16, (95% CI 1.57–2.96), and HR = 2, (95% CI 1.23–3.24); *p* < 0.01 for all three comparisons], in the adjuvant setting there was not a statistically significant difference in terms of OS, PFS, LCR and DM [85].

The radiotherapy Gy dose for OPSCC adjuvant treatment is a significant topic of debate, considering that usually the standard method provides for a 60 Gy dose, and it is associated with a higher risk of dysphagia [88,89,90].

By contrast, some authors distinguished the standard use of adjuvant chemotherapy according to different extracapsular extension (ECE) subtypes in p16+ patients. Patient treatment was tailored according to risk classes identified during diagnostic assessment, in particular involving radiation therapy only, without cisplatin in lower risk cases [91].

The role of de-intensified adjuvant chemo-radiotherapy was assessed by the SIRS TRIAL [89]. The authors included TORS-treated patients and distinguished them according to various prognostic features and assigned into three groups: no risk features, intermediate risk factors (peri-neural and/or lympho-vascular invasion) and high-risk factors (ENE positive, more than three positive lymph-nodes and/or positive margin). In the presence of intermediate risk factors has been realized a de-intensification radiotherapy (50 Gy). After a median follow-up of 43.9 months, the authors reported a DSS comparable between the de-escalated radiotherapy group and high-risk group, supporting the decreased dose radiation for HPV-related OPSCC undergoing TORS.

Recently, De Virgilio et al. [55] analyzed recent and future de-intensification strategies in the treatment of oropharyngeal carcinomas. They reported that several de-intensified approaches have been published, with the aim of providing patients with less toxic treatments while maintaining comparable results in terms of disease control and survival. The authors concluded that considering the conflicting results reported so far by preliminary studies, it is necessary to wait for the final results of on-going trials to identify the best de-intensification strategy and which patients would really benefit from it.

### 3.5. Salvage Surgical Treatment

Several authors reported high recurrence rates (about 50%) in HPV-negative patients with stage III and stage IV OPSCC [92,93].

Salvage surgery remains the only curative option for patients with local treatment failure after primary radiation or systemic treatment [94,95,96].

However, there is limited data on surgical salvage success rates for OPSCC related to human papillomavirus and p-16 status.

In order to analyze the role of salvage surgery for locally recurrent or persistent oropharyngeal tumours, Patel et al. evaluated thirty-four patients treated with chemoradiotherapy [95]. Although the authors found surgical salvage to be a feasible approach, especially in patients without regional recurrence or with an achievable clear margin, p16 status did not result in a prognostic impact for local recurrence rates (*p* = 0.06).

However, other authors have reported OS improvements after surgical salvage at locoregional and DM control both in HPV-positive and HPV-negative [91,93] patients. 

A retrospective study including 65 loco-regional and 43 distant metastatic patients found surgical salvage associated with better overall survival rates after disease recurrence (HR, 0.26; *p* = 0.002) [79].

Furthermore, a recent retrospective analysis of 102 patients reported better 5-year OS rates in OPSCC HPV-related after salvage surgery (HR: 0.34); conversely, positive margins expressed a poor prognosis (HR: 2.65) [22].

Among surgical approaches, TORS has been proven to be a valid alternative to open surgical procedures in the treatment of recurrent oropharynx tumours. 

White et al. reported a significantly positive margin reduction in TORS patients compared to open approach subjects (*p* = 0.007) and a higher 2-year recurrence-free survival rate (74% vs. 43%; *p* = 0.01). The functional outcomes were proven to be more favourable too, with a diminished length of hospital stay (8.0 vs 3.8, *p* < 0.001), rate of tracheostomy (79% vs. 23%, *p* < 0.001) and need for a feeding tube (75% vs. 35%, *p* < 0.001) [96].

### 3.6. Surgical Limits and Contraindications

Surgical treatment in advanced OPSCC is burdened by the peri/post-operative complications of open approach surgery that may require bone resections and reconstructions with pedunculated or free flaps, also affecting the length of hospitalization.

A retrospective analysis by Santoro et al. reported that 43.4% of patients undergoing surgery for oral/oropharyngeal carcinomas experienced post-operative complications (systemic, local, or both) [97,98]. Among them, pulmonary disorders (12%) were the most common systemic complications, followed by generalized infection (9%) and cardiovascular problems (7.6%). Instead, minor complications such as bleeding (7%), flap necrosis (6%), and fistula (5%) were frequent, in line with other studies in literature [98,99].

Moreover, patients undergoing transoral treatment for oropharyngeal tumours present a higher risk of dysphagia and aspiration and may require percutaneous gastrostomy for feeding [100,101]. All these factors can negatively affect the patient’s quality of life.

Although transoral approaches significantly reduce these disadvantages compared to traditional open surgery, literature data comparing treatment approaches for advanced-stage tumours are limited [75,102,103,104].

Gross’s study has shown that OPSCC pT4 patients were more likely to have worse swallow function than pT3 ones, in line with other previous studies [104]. The study also showed that the risk of dysphagia is three times higher in patients undergoing surgery with ≥50% resection of the base of tongue (BOT).

Advances in reconstruction, usually through the microvascular flap, have improved functional capacity for patients who undergo surgery [105,106]. Netscher et al. demonstrated that patients undergoing free flap reconstruction for advanced OPSCC surpassed their pre-treatment functional capacity one year after treatment [105].

Otherwise, TORS may be contraindicated in cases of limited mouth opening, prevertebral fascia involvement or where the resection involves a >50% resection of the BOT or the posterior pharyngeal wall, tumours located on the midline of the BOT or vallecula and neoplastic involvement or extreme proximity of the neoplasm to the internal carotid artery [107].

## 3.7. Checkpoint-Inhibitors Therapy

Checkpoint-inhibitors have been indicated as a possible treatment in case of OPSCC recurrences; studies for neoadjuvant treatment options are ongoing.

Immune checkpoint blockade exerts profound anti-tumour effects in many cancer types; however, the efficacy of immune checkpoint blockades is greatly affected by the tumour microenvironment.

Clinical trials have demonstrated promising clinical efficacy of anti-programmed death-1 (PD-1) therapy in HNSCCs, and nivolumab and pembrolizumab have been approved for HNSCC refractory to platinum-based therapy. However, the response rates of these immunotherapies are relatively low (13–16%), and progression-free survival is limited in most patients. Moreover, the presence of programmed death-ligand 1 (PD-L1) on tumour cells did not satisfactorily predict response, with 27% of PD-L1+ patients vs. 12% of PD-L1− patients responding.

The unsatisfactory efficacy of anti-PD-1 therapy necessitates the development of predictive biomarkers for patient selection and for effective combination immunotherapy [108,109].

## 4. Conclusions

In advanced OPSCC there is a significant difference in DSS between N0 and N+, primary surgical and primary nonsurgical treatment, and perinodal invasion. P16-negative patients showed a worse DSS than p16-positive patients but reported better outcomes to primary surgery than to nonsurgical treatment. Multivariate analysis identified the N category as an independent prognostic factor for survival [89]. The introduction of progressively less invasive surgical methods, recommended for increasing numbers of cancer patients, is reinforcing the role of transoral surgical approaches. Robotic surgery or open approaches with reconstructive flaps mean that patients can be treated even at an advanced stage, resulting in the de-intensification of subsequent systemic therapy and fewer related side effects. Furthermore, in the event of primary failure of systemic therapy or disease recurrence, the surgical approach constitutes an additional therapeutic option which lengthens patient survival functions. The complexity of therapeutic options, grounded in a “shared decision-making process” requires them to be implemented by a multidisciplinary head and neck team consisting of oncologists, otolaryngologists, reconstructive/plastic surgeons, maxillofacial surgeons, dentists, radiotherapists, and pathologists. 

## Figures and Tables

**Figure 1 medicina-59-00304-f001:**
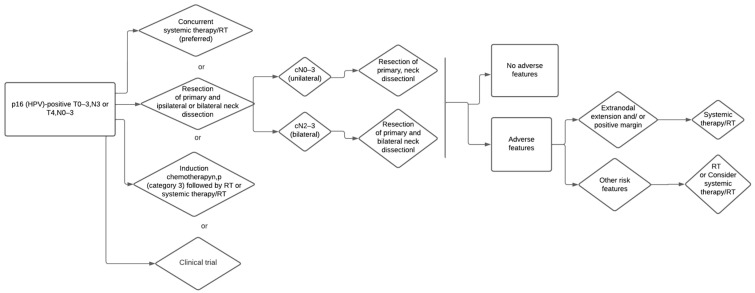
Decisional tree p-16 positive, T0-3 or T4 N0-3.

**Figure 2 medicina-59-00304-f002:**
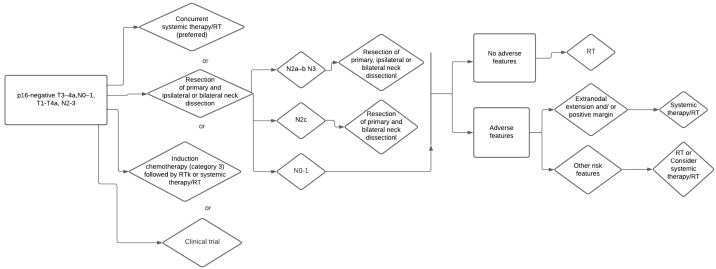
AJCC Decisional tree p-16 negative, T3-T4a N0-1 or T1-4 N2-3.

## Data Availability

Not applicable.

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
