# Peer review of "Surgical Treatment for Advanced Oropharyngeal Cancer: A Narrative Review"

_medicina, 2023, doi:10.3390/medicina59020304_

Round 1
Reviewer 1 Report
The paper carried out a review of literature to analyze different indications of the primary surgical approach, or salvage options after previous systemic treatment failure.
The paper is clear in logic and rich in content, but there are still some limitations:
1. Lack of detailed review of surgical and reconstruction methods.
2. It is better to supplement the rerview of OPSCC diagnosis and typing.
Author Response
All the Authors are grateful for the reviewer's comments
1) the main focus of the Authors would be the critical analysis of surgical treatment for advanced OPSCC, not on the specific surgical techniques, which are distinguished in "open technique" conventional vs transoral.
2) the present paper is part of a special issue on OPSCC and a manuscript, specific on diagnosis and typing will be submitted shortly.
Reviewer 2 Report
Considering this manuscript as an overview of OPSCC, it looks well organized, however, the authors clarify the subject by titling it "Surgical treatment for locally advanced OPSCC". From the viewpoint of "Surgical treatment for locally advanced OPSCC", the article contains confusion and does not seem to provide any information beyond the general guidelines such as NCCN, especially with regard to surgery, which is supposed to be the main focus.
You know today's AJCC classification, OPSCC includes two categories: p16-positive or p16-negative. These should be considered separate and are complicated when considering them at the same time. To simplify the story, it may help define what is "Locally Advanced".
For example, T1 and T2 cancer is a good indication for the transoral surgery you described, is this "Locally advanced"? Is T1-2 p16-positive OPSCC with 5cm of hemilateral lymph node metastasis "Locally Advanced"? T3-4 p16-negative OPSCC sounds "Locally Advanced", how about T1-2 p16-negative OPSCC N1 (Stage III)?
Author Response
All the Authors are grateful for the reviewer's comments
The paper contains different sessions about surgery, distinguishing between HPV16 positive and HPV 16 negative OPSCC patients, as reported in the following session titles:
3.1. Overall Surgical Strategies
3.2. Primary Surgery for HPV/p16 positive
3.3. Primary surgery for HPV/p16 negative
3.4. De-intensification for adjuvant treatment
3.5. Salvage Surgical Treatment
3.6. Surgical limits and contraindications
The main focus of the present narrative review would be an analysis of the surgical treatment in the advanced stages of OPSCC. We agree that the term "locally" could be confusing, and we delete it from the title and the text.
Furthermore, this article would be one of the other manuscripts of a special issue, where there is already published a paper on the early stages of OPSCC.
Round 2
Reviewer 2 Report
Thank you for your reply.
I found my misunderstanding after I sent the initial review, so I sent an email with additional comments to the editorial office as follows, couldn't reach out.
A quick look at the revised manuscript indicates that the typographical error in Figure 1 has been corrected.
In addition, your modification deleting "locally" looks favorable.
1), T1-2N0, is not mandatory to be corrected, as it may be necessary to contrast with advanced cancer.
2) is also acceptable as it is if the authors need it for the structure of the paper.
Please consider 3).
-------------
Additional comment
Authors have defined the materials as OPC stage III and IV, so my previous remark looks off-target. I would like to change my decision into major revision, with following comments.
1) In the primary surgery HPVpositive section.
T1-2N0 is early stage, may not necessary to be included.
2) About Checkpoint inhibitor
The paragraph of Checkpoint inhibitor looks no relation with surgery, is this paragraph mandatory to lead the conclusion?
3) multidisciplinary head and neck team
I could not find any mention of multidisciplinary approach through the article, which seems abrupt. In general, multidisciplinary approach is thought to improve the prognosis of advanced head and neck cancer, so it would be beneficial to mention with new paragraph about it in the manuscript.
There are some typological errors, eg. Figure1
Thanks,
Author Response
Thank you for your update.
2) the paragraph about check-point inhibitors therapy has been added after a suggestion by a previous reviewer, and it highlights the importance of a multimodality approach in the treatment of advanced OPSCC
3) the importance of a multidisciplinary head and neck team is inferable along the text, where there is a comparison between surgery, chemotherapy, and radiotherapy in OPSCC. Furthermore, the importance of an MDT is pointed out at the end of the concluding session.